# Insights into Adaptive Regulation of the Leaf-Petiole System: Strategies for Survival of Water Lily Plants under Salt Stress

**DOI:** 10.3390/ijms24065605

**Published:** 2023-03-15

**Authors:** Xiaojing Liu, Shaozhou Chen, Fengfeng Du, Linhe Sun, Qianhao Huang, Xiaojing Gao, Jinfeng Li, Haiying Tong, Dongrui Yao

**Affiliations:** 1Jiangsu Key Laboratory for the Research and Utilization of Plant Resources, Institute of Botany, Jiangsu Province and Chinese Academy of Sciences (Nanjing Botanical Garden Mem. Sun Yat-Sen), Nanjing 210014, China; 2Jiangsu Engineering Research Center of Aquatic Plant Resources and Water Environment Remediation, Institute of Botany, Jiangsu Province and Chinese Academy of Sciences, Nanjing 210014, China

**Keywords:** water lily, salt stress, adaptive strategies, morpho-physiological regulation, transcriptome

## Abstract

The water lily (*Nymphaea tetragona*) is an ancient angiosperm that belongs to the Nymphaeaceae family. As a rooted floating-leaf plant, water lilies are generally cultivated in fresh water, therefore, little is known about their survival strategies under salt stress. Long-term salt stress causes morphological changes, such as the rapid regeneration of floating leaves and a significant decrease in leaf number and surface area. We demonstrate that salt stress induces toxicity soon after treatment, but plants can adapt by regenerating floating leaves that are photosynthetically active. Transcriptome profiling revealed that ion binding was one of the most-enriched GO terms in leaf-petiole systems under salt stress. Sodium-transporter-related genes were downregulated, whereas K^+^ transporter genes were both up- and downregulated. These results suggest that restricting intracellular Na^+^ importing while maintaining balanced K^+^ homeostasis is an adaptive strategy for tolerating long-term salt stress. ICP-MS analysis identified the petioles and leaves as Na-hyperaccumulators, with a maximum content of over 80 g kg^−1^ DW under salt stress. Mapping of the Na-hyperaccumulation trait onto the phylogenetic relationships revealed that water lily plants might have a long evolutionary history from ancient marine plants, or may have undergone historical ecological events from salt to fresh water. Ammonium transporter genes involved in nitrogen metabolism were downregulated, whereas NO_3_^−^-related transporters were upregulated in both the leaves and petioles, suggesting a selective bias toward NO_3_^−^ uptake under salt stress. The morphological changes we observed may be due to the reduced expression of genes related to auxin signal transduction. In conclusion, the floating leaves and submerged petioles of the water lily use a series of adaptive strategies to survive salt stress. These include the absorption and transport of ions and nutrients from the surrounding environments, and the ability to hyperaccumulate Na^+^. These adaptations may serve as the physiological basis for salt tolerance in water lily plants.

## 1. Introduction

The water lily is a perennial floating plant of the Nymphaeaceae family, which is one of the most ancient angiosperm lineages [1]. Water lilies are famous for their colorful flowers and strong fragrance [2,3]. The water lily flower is also widely used for essential oil extraction and its medicinal properties [4,5]. Most studies of the water lily mainly focus on genetic evolution, flowering rhythm, essential oil extraction, and seed biology. However, very little information exists about the mechanism of water lily salt tolerance in saline waters [1,3,5,6]. Water lilies are generally cultivated in fresh water, which may explain why so few studies have investigated their responses to salt stress. According to a vegetation survey conducted in the northern part of the southeast island of Scotland, the floating leaves of the water lily can survive in water with a salinity of 1.3% [7]. This is evidence that water lilies may have the genetic capacity to tolerate saline environments for extended periods of time. Through field investigation and laboratory analysis, we found that water lilies can be domesticated in saline water in the coastal regions of China. Studies on salt tolerance in water lily plants may lead to the increased use of these plants in the landscaping of coastal regions, and allow saline water to be used for landscape irrigation in coastal cities lacking freshwater.

It is well-known that the growth of land plants can be significantly inhibited by salt stress [8]. Plant biomass can decline rapidly as the salt content in soil increases [9,10]. Not only does salt stress directly inhibit plant growth, but it also inhibits photosynthesis and reduces the synthesis of growth hormones. The effect of salt stress on photosynthesis is mainly due to a reduction in leaf surface area, chlorophyll content, stomatal opening, and to some extent, photosystem II efficiency [11,12,13]. Salt stress generally affects plants by causing the accumulation of toxic ions, osmotic stress, unbalanced nutrient uptake, and oxidative stress [14]. Studies have also demonstrated the importance of metabolic processes in plants under salt stress. In fact, disruption of the nitrogen metabolism pathway is the main cause of salt damage to plants [15].

Land plants have evolved multiple strategies to cope with salt stress. The roots transmit salt stress signals to the shoot and maintain ion balance by regulating signal transduction and ion-transport-related proteins [16,17]. The aboveground organs regulate photosynthesis, carbon and energy metabolism, and reactive oxygen species (ROS) scavenging to avoid excessive ion damage [18,19,20]. While this strategy works for land plants, aquatic plants such as water lilies must deal with salt stress through different means. As a rooted floating-leaf plant, the petioles of water lilies are wholly immersed in salt water, and the floating leaves have constant direct contact with salt water. It is thought that aquatic plants evolved a unique ability to absorb ions through their leaves and petioles [21]. However, little is known about the survival strategies used by water lilies growing in saline water.

In the present study, it was proposed that water lilies have an adapted leaf-petiole system that allows them to tolerate being submerged in saline aquatic environments. We investigated the morpho-physiological characteristics, transcriptional regulation, and accumulation of sodium and potassium ions in the leaf-petiole system of water lily plants under salt stress. The results aimed to provide reference information for future studies on salt tolerance in water lilies and a wide range of other aquatic plants.

## 2. Results

### 2.1. Phenotypes of Water Lily Plants under Salt Stress

*Nymphaea* ‘Colorado’ water lily plants were treated with 0, 50, 100, 150, and 200 mM NaCl for 30 days. Symptoms of salt stress appeared in plants treated with 50 mM NaCl with severity increasing as the concentration of NaCl increased. Salt stress greatly inhibited the growth of water lily plants. The number of floating leaves decreased, with leaf chlorosis and decay after salt treatment. The images demonstrate a rapid reduction in biomass and the number of floating leaves in plants treated with as little as 50 mM NaCl. Decaying mature leaves and the regeneration of young floating leaves are clearly seen in plants treated with NaCl after 30 days of treatment (Figure 1). Leaf surface area, leaf number, and plant biomass were all negatively correlated with the concentration of NaCl after 30 days of continuous treatment (Figure 2). Quantification of the average leaf area, total leaf area, and average number of leaves confirmed that these parameters declined as the salt concentration increased (Figure 2a–c).

With reductions in leaf area and leaf number likely limiting the photosynthetic capacity of salt-treated water lilies, we anticipated reductions in plant fresh weight (FW) and dry weight (DW). The results confirmed that the FW and DW of petioles and floating leaves were also negatively correlated with the severity of salt stress (Figure 2d). The effects were apparent in as little as 50 mM NaCl with reductions in FW and DW of 44.82% and 36.75%, respectively, relative to the control plants. Plants treated with 200 mM NaCl exhibited 94.75% and 96.20% reductions in FW and DW, respectively, compared with the control plants.

### 2.2. Effects of Salt Stress on Photosynthesis in Water Lily Plants

Multiple photosynthetic parameters were recorded at three-day intervals over a 30-day period in plants treated with 0, 50, 100, 150, and 200 mM NaCl. A rapid decline in the leaf net photosynthetic rate (*Pn*) occurred by day 3 for all treatments, and the decline was greater with increasing salt concentration (Figure 3a). After 30 days, the 50 and 100 mM NaCl treatments resulted in a *Pn* decrease of 17.69% and 49.49%, respectively, relative to the control. At higher concentrations of 150 and 200 mM NaCl, *Pn* decreased by 62.01% and 82.62%, respectively, relative to the negative control. Notably, the *Pn* stabilized after day 6 for plants treated with 50 mM NaCl. This stabilization effect was also observed at higher salt concentrations but occurred after 12 to 15 days, and the stabilized *Pn* values were reduced by approximately 50% or more compared to the negative control. This stabilization in the *Pn* suggests that water lilies can tolerate photosynthetic damages caused by salt stress, and indicates that increased photosynthetic damage likely occurs when plants are subjected to salt concentrations greater than 50 mM NaCl.

The intercellular CO_2_ concentration (*Ci*) increased initially, but was followed by relative stability in the time after salt treatment (Figure 3b). However, no significant changes were observed in the transpiration rate (*Tr*) or stomatal conductance (*Gs*) between treatments at each time point (Figure 3c,d). As a typical floating plant, water lilies have no stomata in their lower epidermis, so stomatal openings are only distributed in the upper epidermis of mature leaves (Appendix A). Therefore, we speculate that the decline in *Pn* induced by salt stress depended mainly on non-stomatal factors.

To further determine the effects of salt stress on water lily plants, chlorophyll fluorescence parameters of floating leaves were measured. A rapid decline in the photosystem II (PSII) actual photochemical quantum yield (Φ_PSII_), electron transport rate (ETR), and photochemical quenching coefficient (*qP*) occurred within the first nine days after salt stress treatment (Figure 4b–d). These results indicated that salt stress may decrease the rate of photochemical reactions in the floating leaves, and the energy absorbed by chlorophyll in PSII may dissipate through heat and fluorescence. In leaves treated with 50 and 100 mM NaCl, a near-complete recovery of chlorophyll fluorescence parameters was achieved after 18 days. This result suggests that the chlorophyll in water lily leaves can eventually acclimate to NaCl concentrations of 100 mM or less. At NaCl concentrations higher than 150 mM, water lily plants could survive, maintain continuous growth, and regenerate young floating leaves but suffer from more severe and prolonged effects on photosynthesis.

### 2.3. RNA-Seq Analysis of Leaves and Petioles from Water Lily Plants Treated with NaCl

We used the Illumina NovaSeq 6000 system to perform RNA-seq on young floating leaves and petioles from *N.* ‘Colorado’ water lily plants treated with 150 mM NaCl. The goal of this experiment was to investigate the gene networks under long-term salt stress acclimation in water lilies. The sequenced cDNA libraries generated between 20.4 and 24.0 million raw reads. The total number of clean reads per library ranged from 19.8 to 23.7 million after removing poly-N reads, adapters, and low-quality reads (Table 1). The length distributions of unigenes are available in Appendix A. The clean reads were assembled into transcripts using Trinity software (v2.4.0). The transcripts were mapped to the NCBI RefSeq database, and the mapping rate range from 66.72% to 69.92% (Appendix A). The sequences were compared to the Nr, Nt, Pfam, KOG/COG, Swiss-Prot, KO, and GO database and 136,306 genes were annotated (Appendix A). FPKM values were calculated to quantify the differentially expressed unigenes. The FPKM density distribution is shown in Appendix A. Pearson’s correlation coefficient showed a relatively high correlation between biological replicates of the plant samples (Appendix A).

### 2.4. Analysis of Differentially Expressed Genes

Differentially expressed genes (DEGs) were identified using DEseq by comparing the transcriptomes of the leaves and petioles collected from salt-treated plants to a non-treated control. A total of 2292 and 2355 DEGs were identified in the leaves and petioles, respectively, of water lilies treated with 150 mM NaCl. Of the DEGs, 296 were upregulated in young floating leaves, and 1996 were downregulated. In the petioles, 476 genes were upregulated, and 1879 were downregulated (Figure 5a). Floating leaves and petioles subjected to 150 mM NaCl shared 545 DEGs, whereas 1747 DEGs were leaf-specific, and 1810 DEGs were petiole-specific (Figure 5b).

### 2.5. Significantly Enriched Pathways in Water Lilies under Salt Stress

Gene ontology (GO) and Kyoto Encyclopedia of Genes and Genomes (KEGG) analyses were performed to identify significantly enriched pathways from the DEGs. Genes were categorized using Goseq software with Wallenius’ noncentral hypergeometric distribution, and the GO terms with corrected *p*-values less than 0.05 were considered significantly enriched. Twenty GO terms were identified as significantly enriched with DEGs from floating leaves, and eleven GO terms were identified as significantly enriched with DEGs from petioles (Figure 6a,b). Notably, the ion binding pathway (GO:0043167) was significantly enriched in both the leaves and petioles: 443 DEGs in leaves and 396 DEGs in petioles. This result suggests that ion binding in the leaves and petioles is an important biological process used by water lilies for adaptation to salt stress.

Notable pathways enriched in both leaves and petioles from the KEGG analysis included starch and sucrose metabolism and plant hormone signal transduction (Figure 7a,b). In total, 36 genes involved in starch and sucrose metabolism were downregulated under salt stress. This decrease in carbohydrate metabolism might explain the reduced photosynthetic capacity of plants subjected to salt stress. Genes involved in plant hormone signal transduction included DEGs related to auxin, ethylene, abscisic acid, and gibberellin that were identified in water lily plants under salt stress, and auxin-related genes accounted for more than 50% DEGs.

### 2.6. Ion and Water-Transport-Related Genes in Water lily Plants under Salt Stress

The GO and KEGG analyses highlighted the importance of ion binding in the response to salt stress in water lily leaves and petioles. We decided to take a closer look at DEGs involved in ion absorption and ion transport. A total of 65 DEGs involved in ion transport were identified in both the leaves and petioles. These DEGs included genes encoding sodium transporters, potassium transporters, chloride channels, anion channels, cation transporters, and ABC transporters. Thirty-eight of these DEGs were differentially expressed in the leaves, thirty-four in the petioles, and only six overlapped in both leaves and petioles. Thirty-seven DEGs were downregulated under salt stress. Two differentially expressed sodium transporter genes (Cluster-3509.15636 and Cluster-3509.89148) were downregulated in water lily plants under salt stress. Gene Cluster-3509.15636 was downregulated in both the leaves and petioles, whereas gene Cluster-3509.89148 was specifically downregulated in the petioles (Table 2). Based on these expression data, it seems that water lilies may adapt to salt stress by shutting down sodium transport to restrict excessive sodium from accumulating in the plant. As it was reported that AQPs can mediate ion transport across the membranes, we further identified water-transport-related DEGs in water lily plants. In total, four water-transport-related DEGs were identified in water lily plants. Notably, two aquaporin TIPs (Cluster-3509.66551 and Cluster-3509.74762) were identified as significantly downregulated genes in floating leaves (Table 2).

Since nitrogen metabolism disorder is the main cause of salt damage in plants, we further investigated the expression of nitrogen uptake and transport-related genes. DEGs associated with nitrogen transport and metabolism were identified, including proton-dependent oligopeptide transporter, oligopeptide transporter, vacuolar amino acid transporter, ammonium transporter, NRT1/PTR FAMILY protein, and S-type anion channel SLAH et al. Sixteen putative transporters were differentially expressed in the leaves, seventeen in the petioles, and five in both the leaves and petioles. Genes encoding ammonium transporters were downregulated, whereas genes encoding NO_3_^−^ transporters, including the NRT1/PTR FAMILY protein, were upregulated (Table 2). Both upregulation and downregulation were found among the S-type SLAH family of anion channels in the leaves and petioles, suggesting a selective bias toward NO_3_^−^ under salt stress (Table 2).

### 2.7. Differential Expression of Genes Related to Plant Hormones

Plants have evolved multiple strategies for integrating exogenous salt stress signals into responses that balance growth with salt tolerance. These signals are communicated systemically through the action of different hormones, which play an important role in regulating stress responses and tolerance [22]. Auxin-related genes constitute the largest component of the plant hormone signal transduction pathway. Our RNA-seq analysis identified 19 and 24 auxin-related DEGs in the leaves and petioles, respectively (Table 3). Most of these auxin-related genes were downregulated. The results suggested that downregulation of the auxin response pathway in salt-stressed plants may lead to reduced plant biomass. Water lilies may tolerate salt stress by inhibiting plant growth to shuttle metabolic resources toward a salt adaptation response. Ethylene is another important hormone involved in salt tolerance signaling in water lily plants. We identified five and twelve ethylene-related DEGs in the leaves and petioles, respectively. These results suggest that ethylene may act as a signaling molecule that negatively regulates salt tolerance in water lily plants. Although many reports indicate that ABA can induce adaptive responses to salt stress in maize, rice, *Arabidopsis*, and other species, few DEGs involved in ABA or gibberellin responses were identified in our experiment [23,24,25,26]. These results suggest that ABA and gibberellin may not play an important role in long-term salt adaptation in water lilies. To validate the expression of transcriptome data, several genes were selected for quantitative real-time PCR (qRT-PCR) (Figure 8). Generally, the qRT-PCR results of these genes were consistent with the transcriptome data.

### 2.8. Changes in Sodium (Na) and Potassium (K) Contents in Salt-Stressed Leaves and Petioles

To investigate the Na and K contents in salt-stressed plants, the leaves and petioles were sampled after 150 mM NaCl treatment. The K content of leaves and petioles decreased in salt-stressed plants, with most of the decline occurring in the first three days of treatment (Figure 9). In contrast, the Na content of leaves increased up to day 12 of treatment before stabilizing, whereas the Na content of petioles increased up to day three before stabilizing (Figure 9). As a freshwater plant, water lilies accumulate a large amount of Na in their leaves and petioles, which results in a high Na/K ratio under salt stress. When water lilies were grown under salt stress, the petiole rapidly accumulated Na until saturation occurred around day three at a concentration of 80 g kg^−1^ DW. The Na content peaked within 12 days in the leaves and saturated in the petioles a few days later. Although the transcriptomic data suggest that both the leaves and petioles have an ion exchange capacity, these results highlight the important function of the petioles in Na absorption and transportation in short-term salt stress. 

## 3. Discussion

The root system of terrestrial plants is the first organ to encounter salt stress. Roots are also the first to induce a systemic physiological response through signal transduction, ion absorption, and long-distance ion transport from the roots to the shoots [16,27]. This systemic response requires coordinated gene regulation between the roots and the shoots [28]. As a floating aquatic plant, water lilies have morphologically and physiologically adapted to their aquatic environment. The fact that water lily roots and shoots are both surrounded by water suggests that both organs sense and respond to salt stress simultaneously. In this study, we found that water lilies adapt to salt stress by decreasing the number and surface area of its floating leaves along with the rapid regeneration of new floating leaves (Figure 1 and Figure 2). Photosynthetic data revealed the effects of salt-induced toxicity. A rapid drop in the photosynthetic rate and chlorophyll fluorescence parameters occurred soon after salt treatment. However, water lily plants exposed to lower levels of salt stress were able to partially recover over time (Figure 3 and Figure 4). These results suggest that water lilies can adapt to salt stress by adjusting their morphological and physiological strategies to become salt tolerant (Figure 10). Water lilies have adapted to saline water in a manner that is unique and significantly different from terrestrial plants. In this study, the absorption and transport of ions by the petioles and leaves are typical features used by water lilies to regulate changes in environmental salt concentration. By regulating ion transport across the plasma membrane of stem and leaf cells, water lilies maintain homeostasis between intracellular and extracellular salt concentrations (Figure 10). It had been reported that AQPs can mediate ion transport across the membranes [29,30]. In *Arabidopsis*, AtPIP2;1 is responsible for Na^+^ transport, and Na^+^ uptake and accumulation can be restricted by internalizing AtPIP2;1 from the plasma membrane [31]. AQPs are also reported as turgor sensors to regulate conductance of K^+^ channels [32]. In water lily plants, two aquaporin TIPs genes were identified as significantly downregulated genes in floating leaves (Table 2). Possibly TIPs were involved this biological process, and work together with sodium as osmoregulators in water lily plants.

In this study, the RNA-seq results suggest that most of the differentially expressed genes were organ-specific; only 13.29% DGEs overlapped in the leaves and petioles under salt stress (Figure 5). Further analysis of ion channels and transporters suggested that the petioles and leaves tend to utilize different ion channels and transporters under salt stress (Table 2). In addition, the KEGG pathway analysis showed that the enrichment of the ribosome pathway was only found in the floating leaves under salt stress, but not in the submerged petioles of water lily plants. In *Arabidopsis*, ribosomes are highly heterogenous, and each organ might need a different association of non-paralogous ribosomal proteins. It was reported that the expression of ribosomal protein genes varied dramatically in different organs [33]. In *Brassica napus*, it was also reported that the number of paralogues expressed for each ribosomal protein gene varied extensively with tissue types [34]. In this study, the floating leaves and the submerged petioles of water lily plants might have a distinct population of ribosomal protein gene transcripts, and the physiological state, such as salt-induced osmotic stress, might demand different amounts of ribosomes between organs. Moreover, salt treatment led to leaf chlorosis and decay in floating leaves, which might include the process of cell apoptosis. The decline in chloroplast protein synthesis was associated with loss of polyribosomes, and cell apoptosis also results in the degradation of ribosomes [35,36]. Therefore, the enrichment of the ribosome pathway in floating leaves indicates functional differences in the ways in which leaves and petioles adapt to salt stress.

Sodium is an essential ion that can damage plants at high concentrations [37]. The extracellular concentration of Na directly impacts the intracellular ionic balance and cellular activities that rely on this balance [38,39]. Terrestrial plants have efficient ion transport and selective absorption mechanisms to maintain intracellular homeostasis. Common strategies to avoid cytotoxic levels of Na include selective absorption, efflux, and sequestration of Na [40]. It was previously reported that angiosperm species can be classified into Na-excluders, Na-responders, and Na-accumulators. Species of Caryophyllales raised in environments lacking salt with a shoot Na concentration over 4 g kg^−1^ DW were classified as Na hyperaccumulator species [41]. Coastal plant species with a leaf Na concentration over 30 g kg^−1^ DW were defined as Na hyperaccumulators [42]. Upon salt treatment, Na hyperaccumulation occurred in the leaves and petioles of water lilies within 3 days. While they are generally considered to be freshwater plants, our results suggest water lilies exhibit the characteristics of Na hyperaccumulators. In the absence of salt, the Na content in floating leaves exceeded 15 g kg^−1^ DW, which is higher than that of many terrestrial plants. The Na content in the petioles of water lilies exceeded 40 g kg^−1^ DW, which is higher than that of coastal plants that hyperaccumulate Na. These results suggest that water lily plants naturally hyperaccumulate Na, especially in the petioles. Under salt stress, hyperaccumulation of Na occurred in both the petioles and leaves of water lilies. Three days after the salt treatment commenced, the Na content in the petioles was nearly 80 g kg^−1^ DW, which is almost two-fold greater than that in the petioles of plants raised in fresh water. The Na content of the leaves also rose to 80 g kg^−1^ DW after 12 days of treatment, which was nearly five-fold greater than that in the leaves of plants raised in fresh water (Figure 9). The ability of water lily petioles and leaves to hyperaccumulate Na might be the physiological basis for salt tolerance in water lilies. Water lilies are part of the ANA-grade (*Amborellales*, *Nymphaeales*, and *Austrobaileyales*) angiosperms, and may have retained their genetic potential for salt tolerance from the ancient marine plants that they are descended from.

Plants are often subjected to Na toxicity and K deficiency simultaneously. Sodium itself is antagonistic toward K absorption by cells because Na uptake is linked to K efflux [43,44]. Maintaining intracellular K homeostasis is important for proper plant growth and salt tolerance [45]. Our results demonstrated that the potassium content in the leaves and petioles of water lilies decreased after salt stress, with most of the decline occurring within three days of treatment. After three days, the K content stabilized. It appears that K efflux occurs as an early response to salt stress before the petioles and leaves adapt to prevent further loss of potassium (Figure 9). We hypothesized that the intracellular K concentration may be regulated by potassium-related transporters. Both upregulation (Cluster-3509.83265 and Cluster-3509.8544) and downregulation (Cluster-3509.60348 and Cluster-3509.61628) of potassium transporter genes were observed in the petioles (Table 2). Water lilies may achieve K homeostasis in the petioles to improve salt tolerance. However, the leaves may use a different strategy to prevent K accumulation. Our results showed that three potassium transporter genes were downregulated in the leaves, suggesting that potassium channels may close under salt stress. ABC transporters are also involved in ion channel regulation under salt stress in plants [46,47,48]. In *A. thaliana*, a high expression of *AtMRP5* led to an increase in the ratio of Na to K in seedlings by regulating K uptake in roots to counteract salt stress [49]. In this study, 27 ABC transporter genes were differentially expressed under salt stress. Most of these transporters belong to the ABCC and ABCG subfamilies, which play an important role in Na and K uptake and transport under salt stress [50,51,52]. Our results suggest that water lilies maintain intracellular K above a minimal threshold to acquire salt tolerance.

In summary, floating leaves and submerged petioles use a series of adaptive strategies to survive salt stress. The absorption and transport of ions and nutrients from the surrounding environment are part of this complex biological process. The ability to hyperaccumulate Na may be the physiological basis for salt tolerance in this aquatic plant.

## 4. Materials and Methods

### 4.1. Plant Materials and NaCl Treatment

The water lily cultivar *N.* ‘Colorado’ was obtained from the Nanjing Yileen Garden company. Plants were cultivated in the greenhouse at the Institute of Botany in Jiangsu province and the Chinese Academy of Science. Each tuber was grown in a pot with pond sludge. Then, potted water lily plants were cultivated in a pond. Plants with 5–6 floating leaves were used as experimental materials. For the control plants, potted water lily plants were transferred into deionized water. For salt stress, plants were treated with 50, 100, 150, and 200 mM NaCl, respectively.

### 4.2. Plant Biomass Measurements

After a month-long salt stress treatment, images of water lily plants were taken using a camera (Canon, Melville, NY, USA, EOS 70D). Every floating leaf was scanned using a ScanMaker i800 plus (MICROTEK, Atlanta, GA, USA) scanner. Adobe Photoshop 2022 was used to calculate the leaf area. The fresh weight of the petioles and floating leaves was measured using an electronic balance (Sartorius, Göttingen, Germany, BSA223S). The samples were dried in an oven at 105 °C for 30 min and then dried at 72 °C until no more weight was lost, to measure the dry weight.

### 4.3. Measurement of Gas Exchange and Chlorophyll Fluorescence Parameters

A Li−6800 Portable Photosynthesis System (LICOR Inc., Lincoln, NE, USA) was used to determine gas exchange parameters and chlorophyll fluorescence parameters from 8:00 am to 11:00 am [53]. Each treatment contained ten biological replicates. The concentration of carbon dioxide used in the measurement was 400 mg/L, the light intensity was 1200 μmol/(m^2^·s), and the measurement area was 2 cm^2^. Gas exchange parameters included: net photosynthetic rate (*Pn*), intercellular CO_2_ concentration (*Ci*), transpiration rate (*Tr*), and stomatal conductance (*Gs*).

After 30 min of dark adaptation, the minimum fluorescence *Fo* was recorded. A rectangular flash of 8000 μmol/(m^2^·s) fluorescence was used for excitation and the maximum fluorescence *Fm* was recorded from dark adapted plants. Actinic light of 1200 μmol/(m^2^·s) was used to record *Fm* and steady-state fluorescence *Fs* under light adaptation. Chlorophyll fluorescence parameters, including maximum quantum efficiency of photosystem PSII photochemistry (F_v_/F_m_), PSII actual photochemical quantum yield (Φ_PSII_), electron transport rate (ETR), and photochemical quenching coefficient (*qP*) were calculated.

### 4.4. Transcriptome Analysis of Petioles and Floating Leaves

The leaves and petioles treated with 150 mM NaCl for 3 weeks were collected for RNA-seq analysis. Water lilies raised in deionized water were included as a control. Each treatment had three biological replicates. After collection, the samples were immediately frozen in liquid nitrogen and stored at −80 °C. The total RNA was isolated using an RNAprep Pure Plant Kit (Tiangen, Beijing, China) according to the manufacturer’s instructions.

Purified mRNA was used for library construction. The qualified libraries were sequenced using an Illumina NovaSeq 6000 system. Raw reads in fastq format were cleaned using in-house Perl scripts. Raw data were filtered using fastp (Version 0.19.7). Clean reads were obtained by removing reads containing adapters, N bases and low-quality reads. Q20, Q30, and GC content were calculated. The clean reads were assembled into transcripts using Trinity software (v2.4.0). The transcripts were matched to the NCBI RefSeq database. Gene function was annotated based on Nr, Nt, Pfam, KOG/COG, Swiss-Prot, KO, and GO. FPKM (Fragments Per Kilobase of transcript sequence per Million base pairs) values were calculated to quantify the differentially expressed unigenes. Differential gene expression analysis was performed using the DESeq2 R package (1.20.0). Goseq (1.10.0) and KOBAS (v2.0.12) software were used for GO enrichment analysis and KEGG pathway enrichment analysis of differentially expressed gene sets [54].

### 4.5. ICP-MS Analysis of Sodium (Na) and Potassium (K) Contents

Water lily petioles and floating leaves were sampled at 0, 3, 12, 21, and 30 days after treatment with 150 mM of NaCl. Fresh samples were dried to a constant weight. For ICP-MS analysis, the samples were digested with a mixture of nitric acid and H_2_O_2_ and examined for sodium (Na) and potassium (K) contents using an inductively coupled plasma mass spectrometer (ICP-MS; Agilent Technologies Co., Ltd., Santa Clara, CA, USA) [55].

### 4.6. Quantitative Real-Time PCR

To validate RNA-seq data, ion-transport-related genes Cluster−3509.57758, Cluster−3509.35440, Cluster−3509.102694, Cluster−3509.89148, and Cluster−3509.97678, and auxin-related genes Cluster−3509.33603 and Cluster−3509.66285 were used as candidate genes. The total RNA was extracted using a FastPure Universal Plant Total RNA Isolation Kit (Vazyme, Nanjing, China). The cDNA was synthesized using a HiScript^®^ II Q RT SuperMix for qPCR (+gDNA wiper) synthesis kit (Vazyme, China). RT-PCR was performed on BIO-RAD CFX-Opus 96 (Bio-Rad, Hercules, CA, USA) using ChamQ Universal SYBR qPCR Master Mix (Vazyme, China) [3]. Three biological replicates were analyzed and actin was used for normalization.

## 5. Conclusions

The mechanism by which water lily plants adapt to salt stress requires morpho-physiological and transcriptional regulation in the leaf-petiole system (Figure 10). This includes the rapid regeneration of floating leaves, a decrease in leaf number and area, and photosynthetic adaptation. Plant hormones, especially auxin, appear to be involved in the morphological adaptation to salt stress. Ion binding in the leaf-petiole system was identified as one of the most enriched pathways under salt stress. Specifically, sodium hyperaccumulation, K homeostasis, and a selective bias toward NO_3_^−^ was observed. We found that the leaf-petiole system functions as a Na hyperaccumulator, suggesting that water lilies might have retained their genetic potential for salt tolerance from the ancient marine plants that they are descended from.

## Figures and Tables

**Figure 1 ijms-24-05605-f001:**
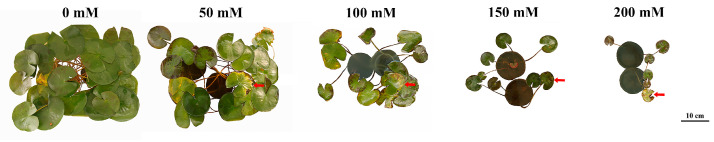
Effect of salt stress on water lily plants. Photos were taken after 30 days of continuous salt treatment. Leaf chlorosis and decay are marked with red arrows.

**Figure 2 ijms-24-05605-f002:**
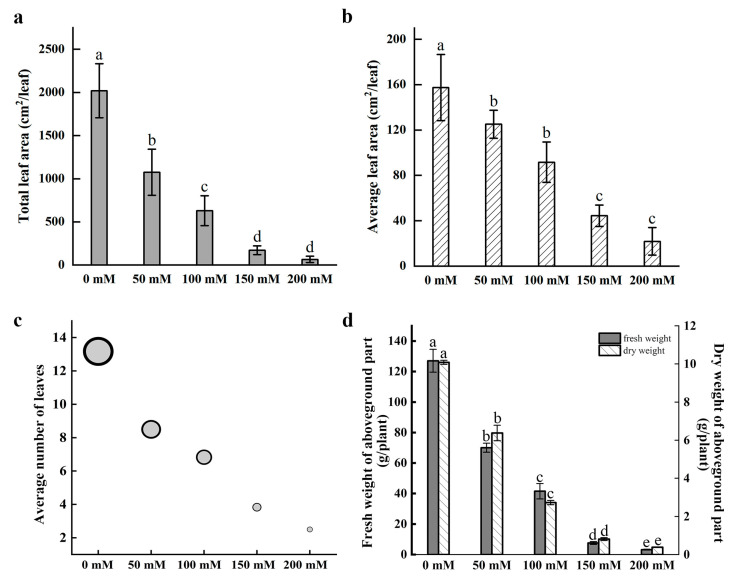
Growth metrics of water lily plants after salt stress treatment. (**a**) Total leaf area per plant. (**b**) Average leaf area per leaf. (**c**) Average number of leaves per plant. (**d**) The fresh weight (FW) and dry weight (DW) of the aboveground parts of plants. Data are the means of three biological replicates (±SDs). Different letters indicate statistically significant differences (*p* < 0.05).

**Figure 3 ijms-24-05605-f003:**
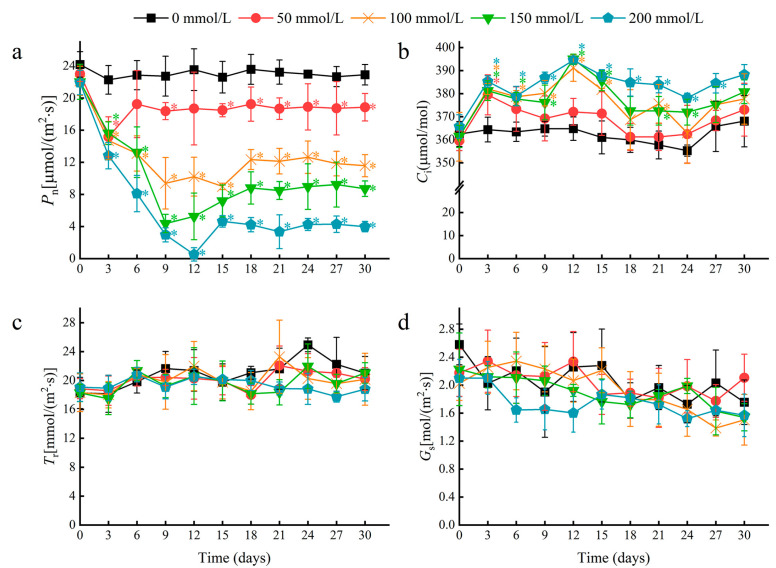
The effects of salt stress on photosynthetic parameters in water lily plants. (**a**) The net photosynthetic rate (*Pn*), (**b**) intercellular CO_2_ concentration (*Ci*), (**c**) transpiration rate (*Tr*), and (**d**) stomatal conductance (*Gs*) in water lily plants subjected to salt stress over 30 days. Data are the means of ten biological replicates (±SDs). The asterisks represent statistically significant differences (*p* < 0.05).

**Figure 4 ijms-24-05605-f004:**
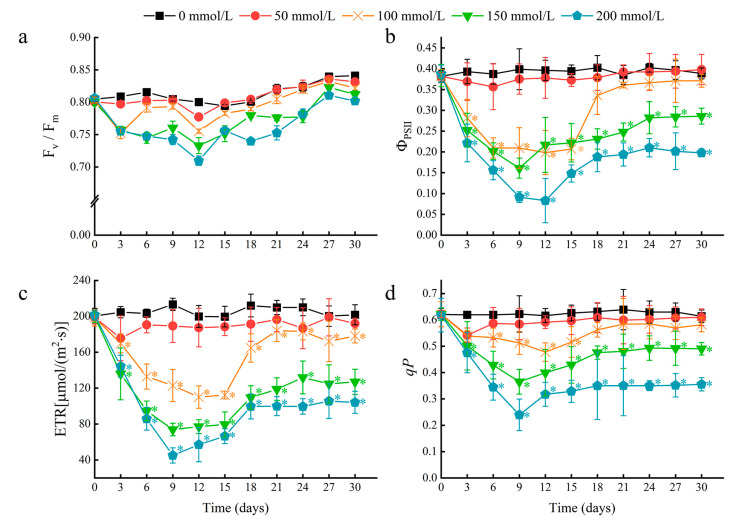
The effects of salt stress on chlorophyll fluorescence parameters in water lily plants. (**a**) Maximum quantum efficiency of photosystem II (PSII) (F_v_/F_m_), (**b**) actual photochemical quantum yield of PSII (ΦPS_II_), (**c**) electron transport rate (ETR), and (**d**) photo-chemical quenching coefficient (*qP*) in water lily plants subjected to salt stress over 30 days. Data are the means of ten biological replicates (±SDs). The asterisks represent statistically significant differences (*p* < 0.05).

**Figure 5 ijms-24-05605-f005:**
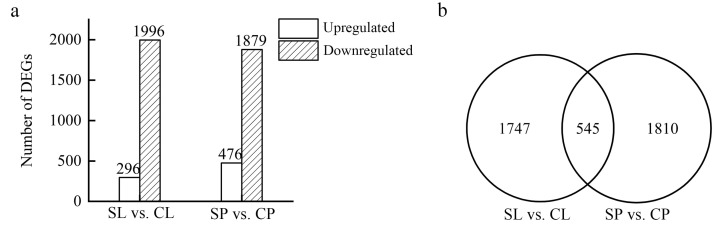
Differentially expressed genes between young floating leaves and petioles under salt treatment. Floating leaves and petioles were collected from water lilies treated with 0 and 150 mM NaCl for 21 days and used for RNAseq: floating leaves treated with 150 mM NaCl (SL), floating leaves treated with 0 mM NaCl (CL), petioles treated with 150 mM NaCl (SP), and petioles treated with 0 mM NaCl (CP). (**a**) Number of DEGs in young floating leaves and petioles under salt treatment. (**b**) Venn diagram of DEGs in young floating leaves and petioles under salt treatment.

**Figure 6 ijms-24-05605-f006:**
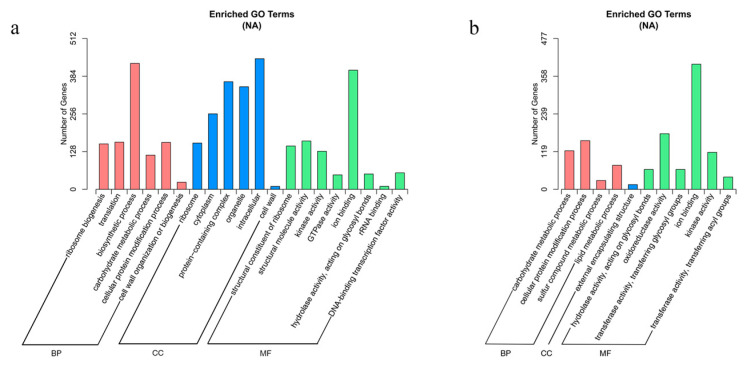
Enriched GO terms based on DEGs identified in leaves and petioles under salt treatment. (**a**) Enriched GO terms in leaves. (**b**) Enriched GO terms in petioles.

**Figure 7 ijms-24-05605-f007:**
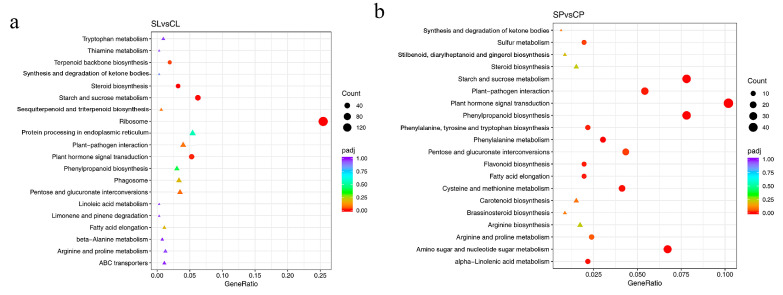
Enriched KEGG pathways based on DEGs identified in leaves and petioles under salt treatment. (**a**) Enriched KEGG pathways in leaves. (**b**) Enriched KEGG pathways in petioles. padj, adjusted *p*-value. Pathways that have a *p*-value under a threshold of 0.05 were identified as significantly enriched pathways, and marked with a circle. Pathways in a triangle have a *p*-value above 0.05.

**Figure 8 ijms-24-05605-f008:**
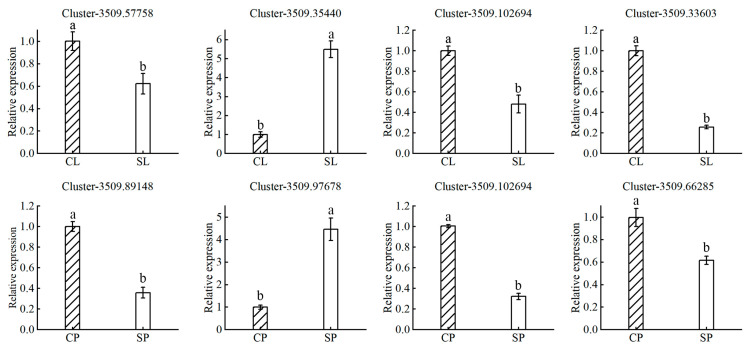
RT-PCR validation of differentially expressed genes. The expression of select genes identified as differentially expressed from the transcriptomic data were validated by RT-PCR in water lily leaves treated with control (CL) and 150 mM NaCl (SL) conditions and petioles treated with control (CP) and 150 mM NaCl (SP) conditions. Actin expression was used for normalization. Data are the means of three biological replicates (±SDs). Different letters indicate statistically significant differences (*p* < 0.05).

**Figure 9 ijms-24-05605-f009:**
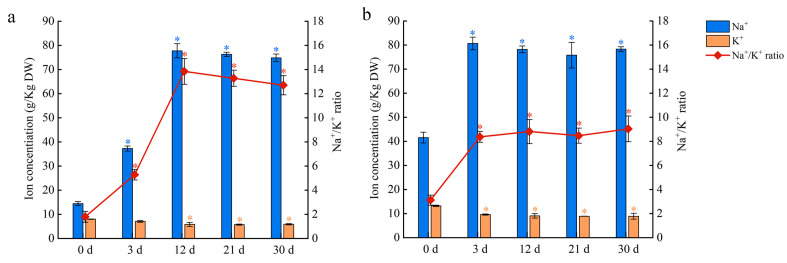
Na and K contents and Na/ K ratio in water lily plants treated with 150 mM NaCl. (**a**) Na content, K content, and Na/ K ratio in leaves. (**b**) Na content, K content, and Na/ K ratio in petioles. Data are the means of three biological replicates (±SDs). The asterisks represent statistically significant differences (*p* < 0.05).

**Figure 10 ijms-24-05605-f010:**
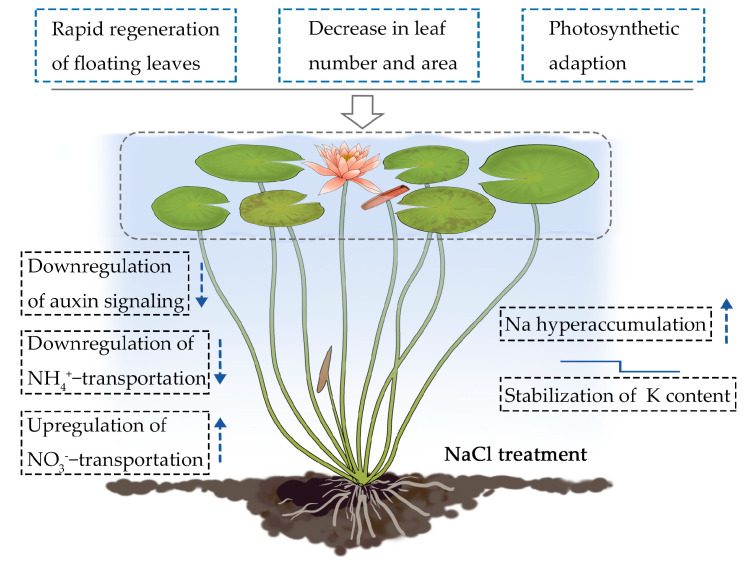
Proposed model for the survival strategies used by the floating leaves and petioles of water lilies under salt stress. The arrow pointing up indicates upregulation of gene expression, and the arrow pointing down indicates downregulation of gene expression under salt stress. The small mark above K content represents minor downregulation followed by homeostasis.

**Table 1 ijms-24-05605-t001:** Sequence statistics of N. ‘Colorado’ under salt stress.

Sample Name	Raw Reads	Clean Reads	Clean Bases	Error Rate (%)	Q20 (%)	Q30 (%)	GC Content (%)
CP1	20,408,879	19,958,021	6.0 G	0.03	97.85	93.75	48.35
CP2	23,177,539	22,564,599	6.8 G	0.03	97.75	93.55	48.10
CP3	21,929,210	21,278,917	6.4 G	0.03	97.67	93.21	48.20
CL1	20,494,403	19,868,591	6.0 G	0.03	97.89	93.81	48.32
CL2	24,003,552	23,703,207	7.1 G	0.03	97.95	93.90	46.17
CL3	22,298,850	20,522,234	6.2 G	0.03	97.94	93.99	48.00
SP1	22,022,491	21,464,315	6.4 G	0.03	97.87	93.77	47.92
SP2	23,673,444	23,099,125	6.9 G	0.03	97.90	93.78	47.23
SP3	21,656,450	21,170,002	6.4 G	0.03	97.76	93.53	47.42
SL1	21,316,148	20,847,247	6.3 G	0.03	97.64	93.08	48.08
SL2	20,647,787	20,119,650	6.0 G	0.03	97.87	93.76	48.04
SL3	23,794,703	23,098,892	6.9 G	0.03	97.89	93.82	46.82

Q20 (%) and Q30 (%): Percentage of bases with Phred quality scores ≥20 and ≥30, respectively. Q20 and Q30 correspond to incorrect base call probabilities of 0.01 and 0.001, respectively. CP, petioles of control plants; CL, leaves of control plants; SP, petioles of salt-treated plants; SL, leaves of salt-treated plants.

**Table 2 ijms-24-05605-t002:** DEGs involved in ion and water transport in the leaves and petioles of water lilies under salt stress.

Gene ID	Leaf		Petiole		Description
	log_2_FC	padj	log_2_FC	padj	
Cluster-3509.63369	2.63	5.53 × 10^−5^	-	-	chloride channel-CLC-g isoform X3
Cluster-3509.15636	−5.53	1.06 × 10^−5^	−8.02	1.30 × 10^−5^	sodium transporter
Cluster-3509.89148	-	-	−3.58	2.02 × 10^−7^	sodium transporter HKT1
Cluster-3509.68931	-	-	1.37	2.09 × 10^−2^	sodium-coupled neutral amino acid transporter 6
Cluster-3509.72547	−1.77	1.53 × 10^−4^	-	-	potassium transporter 5
Cluster-3509.60818	−1.71	7.11 × 10^−4^	-	-	potassium transporter 5
Cluster-3509.37679	−2.11	8.67 × 10^−4^	-	-	potassium transporter 5
Cluster-3509.83265	-	-	1.54	7.46 × 10^−3^	potassium transporter 13
Cluster-3509.60348	-	-	−3.18	9.37 × 10^−3^	potassium transporter
Cluster-3509.8544	-	-	2.75	2.07 × 10^−2^	potassium channel SKOR
Cluster-3509.61628	-	-	−1.45	2.80 × 10^−2^	potassium channel AKT1
Cluster-3509.9398	−8.13	3.34 × 10^−9^	-	-	F-type H^+^-transporting ATPase subunit beta
Cluster-19799.0	−6.60	1.25 × 10^−5^	-	-	F-type H^+^-transporting ATPase subunit alpha
Cluster-3509.35440	8.99	9.46 × 10^−13^	9.51	5.42 × 10^−11^	S-type anion channel SLAH1
Cluster-3509.35441	5.87	1.40 × 10^−9^	5.73	2.71 × 10^−7^	S-type anion channel SLAH1
Cluster-3509.97678	-	-	6.02	6.72 × 10^−3^	S-type anion channel SLAH2-like
Cluster-3509.59617	-	-	−2.50	2.12 × 10^−2^	molybdate-anion transporter
Cluster-3509.72310	−1.82	3.03 × 10^−8^	-	-	Cation-transporting P-type ATPase
Cluster-3509.71994	−1.50	1.43 × 10^−7^	-	-	vacuolar cation/proton exchanger 3
Cluster-3509.9326	−3.46	1.41 × 10^−5^	-	-	Cation transporter HKT6 isoform X1
Cluster-3509.70982	1.59	6.33 × 10^−4^			Cation-transporting P-type ATPase
Cluster-3509.103986	-	-	−3.88	6.33 × 10^−4^	vacuolar cation/proton exchanger 3
Cluster-3509.78160	1.32	6.58 × 10^−6^	-	-	low affinity sulfate transporter 3
Cluster-3509.10275	−6.77	2.47 × 10^−6^	-	-	ABC transporter, ATP-binding protein
Cluster-3509.79575	−6.64	4.14 × 10^−30^	-	-	ABC transporter F family member 4
Cluster-3509.81717	−3.48	4.17 × 10^−4^	-	-	ABC transporter G family member 29
Cluster-3509.72632	−1.89	3.0 × 10^−4^	-	-	ABC transporter G family member 29
Cluster-3509.77443	−1.53	3.31 × 10^−9^	-	-	ABC transporter B family member 9
Cluster-3509.17628	−1.49	2.43 × 10^−7^	-	-	ABC transporter B family member 11
Cluster-3509.60378	−1.37	9.37 × 10^−5^	-	-	ABC transporter C family member 3
Cluster-3509.57758	−1.34	1.57 × 10^−4^	-	-	ABC transporter C family member 3
Cluster-3509.41498	−1.27	6.74 × 10^−7^	-	-	ABC transporter G family member 45
Cluster-3509.65058	1.57	4.07 × 10^−5^	-	-	ABC transporter C family member 10
Cluster-3509.74470	1.61	6.85 × 10^−4^	-	-	ABC transporter C family member 10
Cluster-3509.65059	1.84	1.20 × 10^−7^	-	-	ABC transporter C family member 10
Cluster-3509.98196	1.95	7.41 × 10^−8^	-	-	ABC transporter I family member 17
Cluster-21390.0	3.78	2.93 × 10^−4^	-	-	ABC transporter G family member 6
Cluster-18164.0	6.27	1.15 × 10^−4^	-	-	ABC transporter G family member 6
Cluster-3509.70646	-	-	−4.10	6.78 × 10^−4^	ABC transporter G family member 32
Cluster-3509.100139	-	-	−4.07	1.17 × 10^−3^	ABC transporter G family member 32
Cluster-3509.74471	-	-	1.43	1.54 × 10^−3^	ABC transporter C family member 10
Cluster-3509.84042	-	-	2.78	1.46 × 10^−2^	ABC transporter
Cluster-3509.38354	-	-	−2.14	2.24 × 10^−2^	ABC−1 domain-containing protein
Cluster-3509.49551	-	-	1.65	3.15 × 10^−2^	ABC transporter C family member 10
Cluster-3509.14979	-	-	−3.07	4.49 × 10^−2^	ABC transporter B family member 15
Cluster-3509.45963	-	-	−3.12	4.70 × 10^−2^	ABC transporter G family member 11
Cluster-3509.62876	1.53	1.10 × 10^−5^	1.40	1.88 × 10^−2^	ABC transporter C family member 10
Cluster-3509.65270	1.57	1.09 × 10^−5^	1.32	3.39 × 10^−2^	ABC transporter C family member 10
Cluster-3509.81514	1.81	4.17 × 10^−5^	1.32	4.71 × 10^−2^	ABC transporter C family member 10
Cluster-3509.63514	-	-	2.27	4.00 × 10^−2^	cationic amino acid transporter 2, vacuolar
Cluster-3509.79897	-	-	−4.93	1.20 × 10^−4^	polyamine transporter At3g13620
Cluster-3509.45936	−1.24	1.63 × 10^−2^	-	-	ammonium transporter 1.3
Cluster-3509.57872	−1.16	6.17 × 10^−2^	-	-	ammonium transporter 1 member 1
Cluster-3509.83579	-	-	−3.64	1.66 × 10^−2^	ammonium transporter 2-like protein
Cluster-3509.78326	-	-	−3.87	3.46 × 10^−3^	ammonium transporter 3 member 1
Cluster-3509.45936	−1.24	1.63 × 10^−2^	-	-	ammonium transporter 1.3
Cluster-3509.57872	−1.16	6.17 × 10^−3^	-	-	ammonium transporter 1 member 1
Cluster-3509.83579	-	-	−3.64	1.66 × 10^−2^	ammonium transporter 2-like protein
Cluster-3509.78326	-	-	−3.87	3.46 × 10^−3^	ammonium transporter 3 member 1
Cluster-3509.68250	1.90	1.38 × 10^−3^	-	-	NRT1/ PTR FAMILY 7.3
Cluster-3509.102694	−1.57	1.66 × 10^−7^	−2.78	3.70 × 10^−4^	NRT1/ PTR FAMILY 5.2
Cluster-3509.70042	-	-	1.49	4.70 × 10^−2^	NRT1/ PTR FAMILY 5.10
Cluster-3509.96889	-	-	1.34	1.68 × 10^−2^	NRT1/ PTR FAMILY 6.3
Cluster-3509.49374	-	-	−1.87	9.51 × 10^−3^	NRT1/ PTR FAMILY 6.3
Cluster-3509.63374	-	-	−1.99	3.30 × 10^−3^	high-affinity nitrate transporter-activating protein 2.1
Cluster-3509.66551	−4.90	2.26 × 10^−4^	-	-	aquaporin TIP1−1
Cluster-3509.74762	−3.66	4.24 × 10^−7^	-	-	aquaporin TIP1−1
Cluster-3509.67781	−4.16	3.58 × 10^−6^	−5.08	7.25 × 10^−17^	regulation of transcription; response to water
Cluster-3509.12700	-	-	−4.02	2.49 × 10^−2^	response to water; potassium ion transport

Padj, adjusted *p*-value, FC, fold change.

**Table 3 ijms-24-05605-t003:** Auxin-related DEGs identified in salt-stressed water lily.

Gene ID	Leaf		Petiole		Description
	log_2_FC	padj	log_2_FC	padj	
Cluster-3509.15665	−4.86	1.77 × 10^−18^	-	-	auxin-responsive protein SAUR64
Cluster-3509.16070	−4.45	6.23 × 10^−13^	-	-	auxin-responsive protein SAUR64
Cluster-3509.1089	5.88	3.68 × 10^−4^	-	-	auxin-responsive protein SAUR40
Cluster-3509.18235	−5.24	3.96 × 10^−3^	-	-	auxin-responsive protein SAUR23
Cluster-3509.48442	−2.43	1.22 × 10^−2^	-	-	auxin-responsive protein IAA12
Cluster-3509.24672	−4.85	8.29 × 10^−3^	-	-	auxin-responsive protein SAUR64
Cluster-3509.33603	−5.46	1.25 × 10^−11^	−6.04	3.03 × 10^−5^	auxin-responsive protein SAUR68
Cluster-3509.16069	−5.51	1.11 × 10^−10^	−4.56	8.48 × 10^−4^	auxin-responsive protein SAUR64
Cluster-3509.68953	-	-	−7.01	9.30 × 10^−33^	auxin-responsive GH3 family protein
Cluster-3509.66285	-	-	−4.84	7.91 × 10^−32^	auxin-responsive protein SAUR72
Cluster-3509.18949	-	-	−3.14	1.08 × 10^−5^	auxin-responsive protein SAUR21
Cluster-3509.72331	-	-	−7.03	1.30 × 10^−4^	auxin-responsive protein SAUR50
Cluster-3509.70882	-	-	−1.69	1.81 × 10^−4^	auxin-responsive protein IAA4
Cluster-3509.12053	-	-	−4.98	3.35 × 10^−4^	auxin-responsive protein SAUR50
Cluster-3509.18253	-	-	−2.64	6.98 × 10^−4^	auxin-responsive protein SAUR32
Cluster-3509.66935	-	-	−2.64	7.45 × 10^−4^	auxin-responsive protein SAUR32
Cluster-3509.89040	-	-	−2.82	3.72 × 10^−3^	auxin-responsive protein SAUR71
Cluster-3509.73182	-	-	−1.27	6.51 × 10^−3^	auxin-responsive protein IAA1
Cluster-3509.14808	-	-	−3.58	1.77 × 10^−2^	auxin-responsive protein SAUR50
Cluster-3509.70213	-	-	−3.26	2.15 × 10^−2^	GH3 auxin-responsive promoter
Cluster-3509.104407	−6.95	4.94 × 10^−4^	-	-	Auxin responsive SAUR protein
Cluster-3509.104944	−6.90	5.47 × 10^−4^	-	-	Auxin responsive SAUR protein
Cluster-3509.106362	−4.01	3.91 × 10^−2^	-	-	Auxin responsive SAUR protein
Cluster-3509.52587	-	-	−1.51	8.83 × 10^−3^	auxin response factor 18
Cluster-3509.67431	-	-	2.30	4.21 × 10^−2^	auxin response factor 5-like protein
Cluster-3509.15779	−2.11	5.45 × 10^−3^	-	-	auxin-induced protein 15A
Cluster-3509.60954	−1.11	2.05 × 10^−3^	-	-	auxin-induced protein 22D
Cluster-3509.62463	−2.67	1.32 × 10^−8^	−4.84	6.72 × 10^−12^	auxin-induced protein AUX22
Cluster-3509.38644	−3.86	1.26 × 10^−6^	−8.91	7.49 × 10^−6^	auxin-induced protein AUX22
Cluster-3509.18234	-	-	−4.04	3.93 × 10^−2^	auxin-induced protein 15A
Cluster-3509.69085	−1.34	5.88 × 10^−4^	−1.17	1.97 × 10^−2^	auxin transporter protein 1
Cluster-3509.69832	−1.41	1.60 × 10^−3^	−1.19	6.73 × 10^−3^	auxin transporter-like protein 2
Cluster-3509.29745	−1.94	1.05 × 10^−2^	−1.90	8.04 × 10^−3^	auxin transporter-like protein 2
Cluster-3509.57652	−1.50	3.45 × 10^−2^	−1.28	7.80 × 10^−3^	auxin transporter-like protein 2
Cluster-3509.34017	-	-	−1.13	2.86 × 10^−2^	auxin transporter-like protein 2
Cluster-3509.65534	−1.91	7.46 × 10^−6^	-	-	AUX/IAA8b
Cluster-3509.62206	−2.03	1.77 × 10^−7^	-	-	AUX/IAA protein
Cluster-3509.103850	−2.07	3.00 × 10^−3^	-	-	AUX/IAA protein

## Data Availability

The data that support the findings of this study have been deposited into the CNGB Sequence Archive (CNSA) of China National GeneBank DataBase (CNGBdb) with accession number (CNP0003913).

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
