# Peer review of "Insights into Adaptive Regulation of the Leaf-Petiole System: Strategies for Survival of Water Lily Plants under Salt Stress"

_ijms, 2023, doi:10.3390/ijms24065605_

Round 1

Reviewer 1 Report

This is a reasonably well-structured manuscript.

However, the preprocessing of RNA-seq data is rather scantly described, although it is the critical material of this study. 

A bit more about the quality control of raw reads should be included, and the method for the quantification of transcripts is completely omitted.

Once quantified, multidimensional scaling is commonly performed to examine the relation between samples. Sometimes outliers and/or severe technical biases are identified, which should be very carefully handled in the downstream analyses. However, this part is omitted, too.

Regarding the differential expression analysis, authors seemed to have made the most of the results that are relevant to the study questions, which is OK. However, judging from the GO term and the KEGG pathway enrichment analysis, the whole RNA-seq analysis is questionable, because it is very hard to comprehend that differentially expressed genes are enriched for many human-disease related GO terms and pathways.

Ribosome-related terms are also enriched in the DEGs. This also needs to be carefully investigated to see whether it is a real result or a technical artifact due to incomplete ribosomal RNA depletion.

Regarding the human disease-related GO terms and pathways, unless there is a good reason for them to be there among DEGs, I am concerned about sample contamination. The assessment of mappability of cleaned reads to water-lily reference genome/transcriptome sequence would suggest a quick answer for the contamination concern. Without careful assessment of the library quality (not the sequencing quality), unfortunately, the differential expression analysis cannot have much meaning.

Author Response

Thank you for comments and suggestions.

Q1: The preprocessing of RNA-seq data is rather scantly described, although it is the critical material of this study.

A1: We revised the preprocessing of RNA-seq data. Pleased check line 466-471.

Q2: A bit more about the quality control of raw reads should be included, and the method for the quantification of transcripts is completely omitted.

A2: We added the quality control of raw reads and the method for the quantification of transcripts as suggested. Please check line 466-468 and 469-471.

Q3: Once quantified, multidimensional scaling is commonly performed to examine the relation between samples. Sometimes outliers and/or severe technical biases are identified, which should be very carefully handled in the downstream analyses. However, this part is omitted, too.

A3: To investigate the correlation among samples, we calculated the pearson correlation coefficient within and between groups. The important indicator pearson correlation coefficient showed that the biological repeats in this experiment is reliable and the sample selection is reasonable, R2 between biological duplicate samples ranges from 0.77 to 0.85. No technical biases were found. Above results was shown in Figure S2. Please check the revised supplementary file.

Q4: However, judging from the GO term and the KEGG pathway enrichment analysis, the whole RNA-seq analysis is questionable, because it is very hard to comprehend that differentially expressed genes are enriched for many human-disease related GO terms and pathways.

A4: We checked all the samples and related figures in our lab, and found that the wrong figure of KEGG pathways was uploaded. We apologize for the mistake. To obtain a reliable result of the KEGG pathway, differential gene sets were annotated to the KEGG database again, including animals, plants, fungi, etc. The result showed that the transcriptome data were mapped to many plant species such as Vitis vinifera, Amborella trichopoda, Nelumbo nucifera et al. No human-disease related GO terms and KEGG pathways were found. We revised figure 4. Pleased check line 211.

Q5: Ribosome-related terms are also enriched in the DEGs. This also needs to be carefully investigated to see whether it is a real result or a technical artifact due to incomplete ribosomal RNA depletion.

A5: For the RNA sample preparations, mRNA was purified from total RNA by using poly-T oligo-attached magnetic beads. No ribosomal RNA contamination was found in the samples. The more dynamic ribosomes may enable more rapid responses to salt stress at the translational level. Ribosome-related pathways were also enriched in Hibiscus tiliaceus, wheat and Arabidopsis under salt stress (Yang et al., 2011; Jiang et al., 2017, Stein et al., 2021).

Q6: Regarding the human disease-related GO terms and pathways, unless there is a good reason for them to be there among DEGs, I am concerned about sample contamination. The assessment of mappability of cleaned reads to water-lily reference genome/transcriptome sequence would suggest a quick answer for the contamination concern. Without careful assessment of the library quality (not the sequencing quality), unfortunately, the differential expression analysis cannot have much meaning.

A6: We checked all the samples and related figures in our lab, and found the wrong figure of KEGG pathways was uploaded. We apologize for the mistake. We corrected figure 4. The pathways were all related to plant biological process, cellular component and molecular function. Please check line 211. We mapped transcriptome data of waterlily plants to Nr, Nt, Pfam, KOG/COG, Swiss-Prot, KO and GO database. Many plant species were mapped such as Vitis vinifera (8.1%), Amborella trichopoda (7.7%), Nelumbo nucifera (6.3%), Cinnamomum micranthum (5.3%) et al. Transcriptome data has no relation with human. To ensure the quality of the library, the libraries were quantified by Qubit2.0 Fluorometer, and detected the insert size by Agilent 2100 bioanalyzer. The mapped species are listed as following: 

Reviewer 2 Report

This study examines the physiological responses of an ancient angiosperm (Water lily) to changes in osmotic stress due to exposure to salt.  The authors obtained plants from a garden center and then exposed them to month-long salt treatments at various concentrations. The authors then measured leaf growth, biomass, gas exchange, chlorophyll fluorescence, gene expression, and Na/K levels in leaf vs. petiole tissue.  Overall, plants exposed to higher sodium concentration had differential responses than control plants.  A schematic found in the discussion (Figure 6) summarizes these responses. 

Overall, the paper is written well and provides a new and interesting insight into a rarely studied plant.  I have a 2 few minor comments (and questions) and then 2 majors comments/questions that should be addressed. 

1.  How old were the plants when they were purchased for the study?  Were all the plants the same age?  Were they clones of one another?  This is just to demonstrate that there were no possible variations, age-related or genetic differences in salt tolerance among the plants used in your study. 

2.  I think Figure 6 should be expanded. It would be more clear if text boxes explaining each of the different responses were added with a few words to describe the overall response of water lilies to salt stress.

3.  In the methods, you indicated that you grew the plants in deionized water. Does this mean that the other plants were only grown in deionized water but with salt only?  What was the substate (soil/gravel) used?  Were they detached from roots?  Wouldn't putting plants in deionized water only create a hypo-osmotic environment that is devoid of necessary nutrients (nitrogen, phosphorous, potassium, and other micronutrients)? If plants had access to nutrients by another means, please indicate this in the manuscript, or else you are missing a suitable control in your experiment. 

4.  Why RT-PCR not used to confirm your RNA-Seq results?  This should be done with some of the select genes to confirm gene expression patterns. 

Author Response

Q1. How old were the plants when they were purchased for the study? Were all the plants the same age? Were they clones of one another? This is just to demonstrate that there were no possible variations, age-related or genetic differences in salt tolerance among the plants used in your study.

A1: The tubers of waterlilies were purchased in March, 2020. Then the waterlily plants were vegetative propagated for experiment. In April 2021, plants with 5-6 floating leaves were selected. All plants are in a ‘Colorado’ genetic background. Plant age were consistent. We revised material and methods. Pleased check line 428-433.

Q2: I think Figure 6 should be expanded. It would be more clear if text boxes explaining each of the different responses were added with a few words to describe the overall response of water lilies to salt stress.

A2: We expanded Figure 6 as suggested. Please check line 376.

Q3: In the methods, you indicated that you grew the plants in deionized water. Does this mean that the other plants were only grown in deionized water but with salt only? What was the substate (soil/gravel) used? Were they detached from roots? Wouldn't putting plants in deionized water only create a hypo-osmotic environment that is devoid of necessary nutrients (nitrogen, phosphorous, potassium, and other micronutrients)? If plants had access to nutrients by another means, please indicate this in the manuscript, or else you are missing a suitable control in your experiment.

A3: Each tuber was grown in a pot with pond sludge. Then potted waterlily plants were cultivated in a pond. Plants with 5-6 floating leaves were used as experimental materials. For control plants, potted waterlily plants were transferred into deionized water. For salt treatment plants, potted waterlily plants were transferred into different NaCl solution. The plants were rooted, not detached. Please check line 429-434.

Q4: Why RT-PCR not used to confirm your RNA-Seq results? This should be done with some of the select genes to confirm gene expression patterns.

A4: We added RT-PCR data to confirm RNA-Seq results. Please check line 263-265, 267-272 and 481-489.

Reviewer 3 Report

Dear Authors,

I have a opportunity to review the manuscript entitled: „ Insights into adaptive regulation of leaf-petiole system: strategies for survival of water lily plants under saline conditions” which is considered for publication in IJMS journal. The article is quite interesting and presents some custom condition effects. However at current stage manuscript need a lot of improvements so I could not recommend it for publication. The reason for that decision I present below in a form of list of specific comments:

Minor problems

I strongly suggest to point by point check IJMS publication rules because in major part the reference list is badly prepared. The names of all authors must be mentioned so use et al in IJMS is an error which must be corrected.

Major problems (in many point very serious):

1.       Title section

The use a term saline is methodological error. Saline is a water supply or water region which extremely high levels of salt. The salt levels presented in this work is a lot lower than in truly saline. The precise use will be salt abiotic stress or NaCl treatment stress

2.       Introduction section

This section must ended with precisely formulated aim or/hypothesis of the study. Currently the work has nothing like that which could be precisely named as aim/hypothesis

3.       Results section

I am sorry to say but most of Figures are overloaded with data and also small and low quality. Currently quality level of figures disqualifies article for publication

Figure 1 IS low quality  the statistically markings on charts are barely readable (especially on b and d part). The (a) part presents low small quality of photos in which authors named visual phenotypes. But what is difference in this phenotypes the significant elements must marked on photos to outline the differences on visual part

Figure 2. To many chart on one figure make all elements extremely low quality. Moreover, the significant changes in photosynthesis parameters clearly marked to show significant changes. Currently almost any significance could not be spotted

Figure 4 I so small and extremely low quality only bad pixels coud be seen (again no statistical significance is not marked)

Table 2 and 3 Authors must marked statistically significant changes in DEGs not only changes. Moreover it will be more logical to check firstly the DEGs allocated or connected with water transport/distribution because this is major factor for water-connected plant in any stress

Figure 5 no statistical significant values are marked

4.       Discussion

In my opinion is dramatically short in context of current knowledge about salt stress and also amount of presented data. This make article badly balanced

5.       Material and methods

I do not know how it is possible in modern and ethical science to not make any citation in this part of study. This situation suggest that all used methods was developed from the start by authors. I analyzed whole methodology and in most part the methods are commonly known not developed by authors

Sincerely,

Author Response

Q1: I strongly suggest to point by point check IJMS publication rules because in major part the reference list is badly prepared. The names of all authors must be mentioned so use et al in IJMS is an error which must be corrected.

A1: We have corrected reference list according to IJMS publication rules. Please check Reference section.

Q2:1. Title section. The use a term saline is methodological error. Saline is a water supply or water region which extremely high levels of salt. The salt levels presented in this work is a lot lower than in truly saline. The precise use will be salt abiotic stress or NaCl treatment stress

A2: We used salt stress in the manuscript as suggested. Please check the revised MS.

Q3: 2. Introduction section. This section must ended with precisely formulated aim or/hypothesis of the study. Currently the work has nothing like that which could be precisely named as aim/hypothesis.

A3: We revised introduction section as suggested. Please check line 83-89.

Q4: 3. Results section. I am sorry to say but most of Figures are overloaded with data and also small and low quality. Currently quality level of figures disqualifies article for publication. Figure 1 IS low quality the statistically markings on charts are barely readable (especially on b and d part). The (a) part presents low small quality of photos in which authors named visual phenotypes. But what is difference in this phenotypes the significant elements must marked on photos to outline the differences on visual part

A4: All the figures were provided in a single zip archive on the website of IJMS submission system. All the figures were at a resolution of 300 dpi. We added the statistically markings on Figure 1b and 1d. Please check line 112. Figure 1(a) part are visual phenotypes of waterlily plants, and the photos were taken by camera (Canon, EOS 70D). The difference in the phenotypes were described in the MS. Pleased check line 97-101.

Q5: Figure 2. To many chart on one figure make all elements extremely low quality. Moreover, the significant changes in photosynthesis parameters clearly marked to show significant changes. Currently almost any significance could not be spotted

A5: All the figures were provided in a single zip archive on the website of IJMS submission system. All the figures were at a resolution of 300 dpi. We marked significant changes in Figure 2 as suggested. Pleased check line 153.

Q6: Figure 4 I so small and extremely low quality only bad pixels coud be seen (again no statistical significance is not marked)

A6: All the figures were provided in a single zip archive on the website of IJMS submission system. All the figures were at a resolution of 300 dpi. GOseq and KOBAS software were used for GO function enrichment analysis and KEGG pathway enrichment analysis of differential gene sets. Enrichment analysis based on hypergeometric distribution principle. Differential gene sets are the gene set obtained by significant difference analysis and annotated to the GO or KEGG database. The result of enrichment analysis is the enrichment of all differential gene sets.

Q7: Table 2 and 3 Authors must marked statistically significant changes in DEGs not only changes. Moreover it will be more logical to check firstly the DEGs allocated or connected with water transport/distribution because this is major factor for water-connected plant in any stress

A7: Differential expression analysis was performed using the DESeq2 R package. The resulting P-values were adjusted using the Benjamini and Hochberg’s approach for controlling the false discovery rate. padj<0.05 and |log2(foldchange)| > 1 were set as the threshold for significantly differential expression. Therefore, screening for DEGs is based on statistical significance. DEGs shown in Table 2 and 3 were statistically significant changes. It is a good suggestion to study water transport/distribution for submerged petioles and floating leaves of waterlily plants. However, this study aimed to investigate the morpho-physiological characteristics, transcriptional regulation, and accumulation of sodium and potassium ions in the leaf-petiole system of water lily plants. The topic of water transport/distribution in aquatic plants might be discussed in further research.

Q8: Figure 5 no statistical significant values are marked

A8: We added statistical significant marks in Figure 5. Pleased check line 288.

Q9: 4. Discussion. In my opinion is dramatically short in context of current knowledge about salt stress and also amount of presented data. This make article badly balanced

A9: We revised discussion section. Pleased check line 308-419.

Q10: 5. Material and methods. I do not know how it is possible in modern and ethical science to not make any citation in this part of study. This situation suggest that all used methods was developed from the start by authors. I analyzed whole methodology and in most part the methods are commonly known not developed by authors

A10: Thank you for your suggestion. We checked material and methods section, added citations and references closely related to the MS. Pleased check line 474, 480 ,488 and reference 47-49.

Q11: Extensive editing of English language and style required

A11: The manuscript was polished for English language, grammar, punctuation, spelling and overall style by professional native-language editors at Maximum Academic Press (MAP 6601-23-1-31).

Round 2

Reviewer 1 Report

Authors addressed the concern of potential sample contamination in the response. However, I do not think it was sufficiently done. The authors queried the sequence reads to nr and nt of NCBI and other protein sequence databases, presumably using BLAST or other similar tools. As the sequencing data is from transcriptome, there will be a certain level of similarity to gene sequences (either cDNA or protein) from many other species. So, protein/gene sequences of other closely related plant species would come up to the top of the list, but non-plant species would still show up in the result, which is what was shown in the response. Thus, this result would not be enough to prove that there were no or negligible levels of contamination in the samples. A more proper way would be checking the proportion of reads that are mapped to waterlily reference transcriptome sequences. It should be consistently high across all the samples. 

This is also related to my other comment on the details of RNA-seq data processing. Although the authors have expanded the RNA-seq data processing methods explanation, it was hardly improved. There is still much information about the processing and the quality of transcriptome data to be disclosed. The reference database for the mapping and quantification is essential. The mapping statistics, such as the number or proportion of uniquely mapped, ambiguously mapped and unmapped reads are also essential. After quantification, the distribution of expression levels for each sample is also very useful for assessing the RNA-seq data quality. Typically, one would expect a narrow peak for the genes that are not expressed or expressed at a very low level and a bell-shaped or a similar curve for the remaining genes. These statistics are not necessarily needed in the main text, but they should be available as supplementary information, at least.

Regarding Figure 4, it shows somewhat confusing results. If I read the figure correctly, the ribosome pathway genes account for more than 25% of the DEGs in leaves, which is 4 times or more higher than any other pathway genes. However, the number of genes in ribosome-related GO terms shown in Fig 4a is not particularly high. Even if Figure 4 is all correct, the level of enrichment of ribosome pathway genes is the highest by far, and yet this enrichment is not observed in the petiole-DEGs. 

For all of these points mentioned above, I think that Figure 4 warrants more discussion with the focus on the ribosome. If the enrichment of the ribosome pathway in the leaf-DEGs can be explained by the increased translational activities under salt stress, why are petiole-DEGs not enriched for the ribosome pathways or GO terms? Are there any technical artifacts that might have resulted in the enrichment of the ribosome-related pathway or GO terms in the leaf-DEGs?

Also regarding Figure 4c and 4d, the color-code for the p-value isn’t easy to read to see which pathways have the p-value under the threshold (0.01 or 0.05?). Converting the p-value to log-scale or using different shapes for the pathways above or below the threshold would be helpful.

Reviewer 2 Report

Thank you for addressing my concerns.  The updated paper has made the necessary corrections.

Author Response

Thank you for your comments and suggestions.

Reviewer 3 Report

Dear Authors,

A lot of improvements added but I am sorry again most of Figures are overloaded with data and also small and low quality as it be. Currently quality still is very bad. Figure 1 IS low quality the statistically markings are badly added with errors in part b. It must be checked and corrected. In some chart bars the SD overlap ech other but letter markings indicate statistical significance which is not true at all.  The (a) part presents low small quality of photos in which authors named visual phenotypes. But what is difference in this phenotypes the significant elements must marked on photos to outline the differences on visual part (arrows asterix or something else). Author claimed in their response that that phenotypical changes are described in lines 97-101. In this line not such thing is even present. The phenotypical changes is changes in the looking of leaf like leaf deformation, chlorotic changes or changes in petiole etc. Nothing like that is present.

Figure 2. Problem Still persist: To many chart on one figure make all elements extremely low quality. Moreover, the significant changes in photosynthesis parameters clearly marked to show significant changes. Currently almost any significance could not be spotted.

Figure 4 I is still low quality only bad pixels could be Figure is unreadable

In the case of Table 2 Still it will be more logical to check firstly the DEGs allocated or connected with water transport/distribution because this is major factor for water-connected plant in any stress. Moreover if author claimed that : “aim of the study is investigate the morpho-physiological characteristics, transcriptional regulation, and accumulation of sodium and potassium ions in the leaf-petiole system of water lily plants.” Then the water transportation has enormous impact on accumulation or lack accumulation of sodium and potassium especially membrane channels , auqporins or other elements.

Figure 5 no statistical significant values are marked still. Author claimed that the added the significance markings in version which I have nothing like that is present.

Sincerely,

Round 3

Reviewer 1 Report

I figure that the cleaned reads were assembled to transcripts, and the assembled transcript sequences were compared against gene/protein databases for annotation.

Please make this clear in the manuscript with the details of the workflow.

Also, please double-check typos and grammatical errors such as ‘totally’ in line 411, which should be changed to ‘in total’.

Other than that, I think the responses have addressed my comments/questions adequately.

Author Response

Thank you for comments and suggestions.

Q1: I figure that the cleaned reads were assembled to transcripts, and the assembled transcript sequences were compared against gene/protein databases for annotation. Please make this clear in the manuscript with the details of the workflow.

A1: We improved the details of transcripts assembling and gene function annotation. Pleased check the results section, line 167-172 and materials and methods section, line 448-450.

Q2: Also, please double-check typos and grammatical errors such as ‘totally’ in line 411, which should be changed to ‘in total’.

A2: We carefully checked typos and grammatical errors of the MS, and changed ‘totally’ to ‘in total’ in line 240.

Reviewer 3 Report

Dear Authors,

Manuscript is improved on the good level. I recomend publication.

Sincerely,

Author Response

(The authors gave the same response as above.)
